# RB dependent transcriptional regulation at mitotic centromeres preserves genome stability

Elizabeth A Crowley, Amity L Manning

**Transcripts derived from centromere repeats play a critical role in the localization and activity of kinetochore components during mitosis such that disruption of RNA polymerase II-dependent transcription compromises the fidelity of chromosome segregation. Here, we show that the retinoblastoma tumor suppressor protein (RB), a critical regulator of the G1/S cell cycle transition, additionally plays an important role in the regulation of centromere transcription during mitosis. We find that cells lacking RB experience increased RNA Polymerase II activity at mitotic centromeres and a corresponding increase in nascent RNA transcripts derived from centromere sequences. Together with high levels of centromere transcription and corresponding R-loop formation, RB-deficient cells exhibit centromere DNA breaks and local activation of ATR that correspond with increased centromere localization of Aurora B, destabilization of kinetochore-microtubule attachments, and an increase in anaphase defects. Importantly, reduction of DNA damage, ATR activity, and mitotic defects following inhibition of RNA Pol II, or targeted repression of centromere transcription through centromere tethering of Suv420h2, support that mitotic defects in RB-deficient cells are linked to centromere transcription.**

## Introduction

During cell division newly replicated chromosomes are segregated equally into two daughter cells, ensuring the faithful inheritance of genetic information. This process is dependent on a specialized region of the chromosome known as the centromere (Brinkley & Stubblefield, 1966; Musacchio & Salmon, 2007) which is epigenetically defined by the presence of nucleosomes containing the histone variant Centromere Protein A (CENP-A) (Earnshaw et al, 1986; Palmer et al, 1991; Fachinetti et al, 2013). Proteinaceous structures known as kinetochores are then assembled at each centromere where they govern the interaction between chromosomes and microtubules of the mitotic spindle.

Long thought to be transcriptionally inert, we now appreciate that both the core centromere region and the pericentric flanking regions are actively transcribed by RNA polymerase II (RNAPII) (Wong et al, 2007; Chan et al, 2012; Ideue et al, 2014; Rošić & Erhardt, 2016). Transcriptional activity at centromeres is a conserved property among species (Melters et al, 2013; Talbert et al, 2018) and has an important role in regulating centromere and kinetochore composition (reviewed in Perea-Resa and Blower [2018]). A key function of centromere transcription is to promote chromatin-remodeling that is permissive for loading of CENP-A-containing nucleosomes during the $G_1$ phase of the cell cycle (Jansen et al, 2007; Dunleavy et al, 2011; Bobkov et al, 2018).

However, the transcription of centromeres is not limited to interphase cells and RNAPII is active at mitotic centromeres (Chan et al, 2012). During cell division, centromere transcripts act as molecular tethers that can recruit and activate key centromere and kinetochore proteins, including CENP-C (Politi et al, 2002; McNulty et al, 2017; Bury et al, 2020) which functions to stabilize CENP-A retention (Falk et al, 2015; Guo et al, 2017; McNulty et al, 2017; Watanabe et al, 2019), Aurora B (AurB) kinase, a member of the chromosomal passenger complex (CPC) that modulates kinetochore-microtubule attachments (Ferri et al, 2009; Jambhekar et al, 2014; Quénet & Dalal, 2014; Blower, 2016), and Shugoshin 1, which regulates centromere cohesion (Liu et al, 2015; Chen et al, 2021). The formation of transcription-dependent R loops, where ssDNA is displaced as the transcription machinery moves along the template strand, also recruits the ataxia-telangiectasia and Rad3-related protein kinase (ATR) (Matos et al, 2020). During mitosis, ATR functions, in part, to active Aurora B kinase (AurB) locally at the centromere (Kabeche et al, 2018). Together, ATR and AurB dynamically regulate kinetochore microtubules to promote accurate chromosome segregation during cell division.

The variety of ways in which transcriptional activity at the centromere contributes to mitotic fidelity indicates that genetic or epigenetic changes that alter the accessibility of the centromere to transcription factors or otherwise perturb the level of centromere transcription during mitosis, have the potential to impact both kinetochore composition and the fidelity of chromosome segregation. Consistent with this model, two functionally relevant marks of transcriptionally silent heterochromatin, trimethylation of

Department of Biology and Biotechnology, Worcester Polytechnic Institute, Worcester, MA, USA

Correspondence: almanning@wpi.edu

lysine 9 on Histone H3 (H3K9me3) and trimethylation of lysine 20 on Histone H4 (H4K20me3), are highly enriched at pericentromeric regions and yet are restricted from centromeres (Schotta et al, 2004; Agredo & Kasinski, 2023). Experimental manipulations that permit the spread of heterochromatin into the centromere corrupt deposition of the centromere specific CENPA-containing nucleosomes and lead to chromosome segregation errors (Martins et al, 2020; Sidhwani & Straight, 2023). Interestingly, the absence or loss of heterochromatin from the pericentromere can also compromise the accuracy of chromosome segregation (Martins et al, 2016; Martins et al, 2020; Herlihy et al, 2021).

The retinoblastoma tumor suppressor protein, RB, physically interacts with the enzymes that place the heterochromatic marks H3K9me3 (Suv39) and H4K20me3 (Suv420h2) (Sanidas et al, 2019). RB-dependent recruitment of Suv420h2 is relevant for establishment of pericentric H4K20me3 (Gonzalo & Blasco, 2005; Gonzalo et al, 2005) and loss of RB results in decreased H4K20me3 and cohesin complex (a reader of H4K20me3) at pericentromeres (Gonzalo et al, 2005; Manning et al, 2010, 2014). Here, we demonstrate that high levels of centromere transcription and corresponding mis-regulation of mitotic kinases underlie centromere damage and mitotic errors that result from loss of RB. Furthermore, we show that suppression of mitotic transcription, centromere-targeted restoration of epigenetic silencing, or titration of kinase activity is sufficient to restore mitotic fidelity in cells lacking the RB tumor suppressor. Together these findings indicate that epigenetic regulation of centromeres is a dynamic and targetable process by which to modulate the accuracy of chromosome segregation.

# Results

### RB loss promotes RNA polymerase II-dependent centromere transcription

To determine whether loss of RB impacts transcription during mitosis, we first examined cells with and without RB depletion for evidence of RNA polymerase II (RNAPII) activity. Using human telomerase reverse transcriptase (hTERT)-immortalized retinal pigment epithelial cells (RPE-1) engineered to carry an inducible RB-targeting shRNA construct (hTERT-RPE-1 shRB), we treated cells with 2 μg/ml of doxycycline for 48 h to induce RB depletion (Fig 1A). Nocodazole arrested mitotic cells were collected, and chromosome spreads were prepared for immunofluorescence analysis of total RNAPII and active RNAPII (phosphorylated at Serine 2, which is indicative of elongating polymerase) (Fig 1B). RNAPII activity has previously been described at mitotic centromeres (Chan et al, 2012). Consistent with these reports, hTERT-RPE-1 cells were positive for both total (RNAPII) and active (RNAPII pS2) RNA polymerase II. Using quantitative measures of RNAPII and RNAPII pS2 signal intensity across anti-centromere-antigen (ACA)-labeled kinetochores, we observe that while depletion of RB does not affect the total pool of RNAPII at mitotic centromeres, it does lead to a greater than fourfold increase in RNAPII pS2, indicating that cells lacking RB experience an increase in transcriptional activity at mitotic centromeres (Fig 1C).

To verify that RNA polymerase activity during mitosis corresponds with an increase in the synthesis of centromere transcripts, we first pulsed nocodazole-arrested mitotic cells with 5-ethynyl uridine (5-EU) for 1 h. 5-EU is an analog of uridine that is incorporated during RNA synthesis. Using click chemistry to link EU-labeled RNAs to biotin, followed by streptavidin bead pull down, newly synthesized mitotic RNAs were isolated from bulk cellular RNA (Fig 1D). Quantitative real-time PCR was used to perform comparative analysis of bulk and nascent centromere transcript levels from control and RB depleted mitotic hTERT-RPE-1 cells (Figs 1E and S1A). We find that mitotic cells lacking RB exhibit a significant increase in newly synthesized (nascent) centromere and pericentromere transcripts. We additionally observe an increase in nascent transcription of LINE elements, which are known to be enriched at centromeres. A similar increase in transcript level is seen regardless of whether the sequence of the transcript analyzed is unique to a single chromosome (i.e., D7Z1/2 and D17Z1/b: centromere/pericentromere of chromosome 7 and 17, respectively) or common to multiple chromosomes (i.e., P82H: centromere and K111: pericentromere), suggesting an underlying, wide-spread dysregulation of transcriptional control at mitotic centromeres and pericentromeres in cells lacking RB.

### RB loss promotes DNA damage and ATR activation at mitotic centromeres

Unzipping of the DNA double stranded helix as the RNA polymerase reads and transcribes the nascent RNA leads to the formation of a transient three-stranded DNA-RNA hybrid structure known as an R loop (Thomas et al, 1976; Santos-Pereira & Aguilera, 2015; Hamperl & Cimprich, 2016). To examine R-Loop formation directly, we performed DNA:RNA immunoprecipitation experiments (DRIP) using the DNA:RNA hybrid-recognizing antibody S9.6 or a control IgG. Following 4 h of nocodazole- induced mitotic arrest in hTERT-RPE-1 cells with and without shRB-induced RB depletion, mitotic cells were isolated and DNA:RNA hybrids precipitated. qRT-PCR was then performed to quantify centromere ($α$-SAT1, P82H, D7Z1, D17Z1) and pericentromere (D7Z2, D17Z1b)-derived R-loops. Using this assay, we find an increase in both centromere and pericentromere sequences in RB-deficient cells compared with control cells (Fig S1B). The S9.6 antibody has been reported to recognize both DNA:RNA hybrids and dsRNA molecules. Therefore, to verify qRT-PCR signal is due to DNA:RNA hybrid-containing R loops, immunoprecipitate experiments were performed in parallel with and without the DNA:RNA hybrid-specific nuclease RNaseH1. Importantly, we find that RNaseH1 treatment reduces qRT-PCR signal in both control and RB-deficient conditions, indicating that the observed increase in immunoprecipitated material from RB-deficient cells reflects an increase in R-loop formation.

Although a normal consequence of RNA transcription, hybrid R-loop structures are prone to both single stranded (ssDNA) and double stranded DNA (dsDNA) breaks (Aguilera & García-Muse, 2012; Cohen et al, 2018). Perturbations in the balance between formation and resolution of R-loops contribute to DNA damage and genomic instability (Costantino & Koshland, 2018). Given our observation that loss of RB increases mitotic centromere transcription and R-loop formation (Figs 1 and S1B), and increased

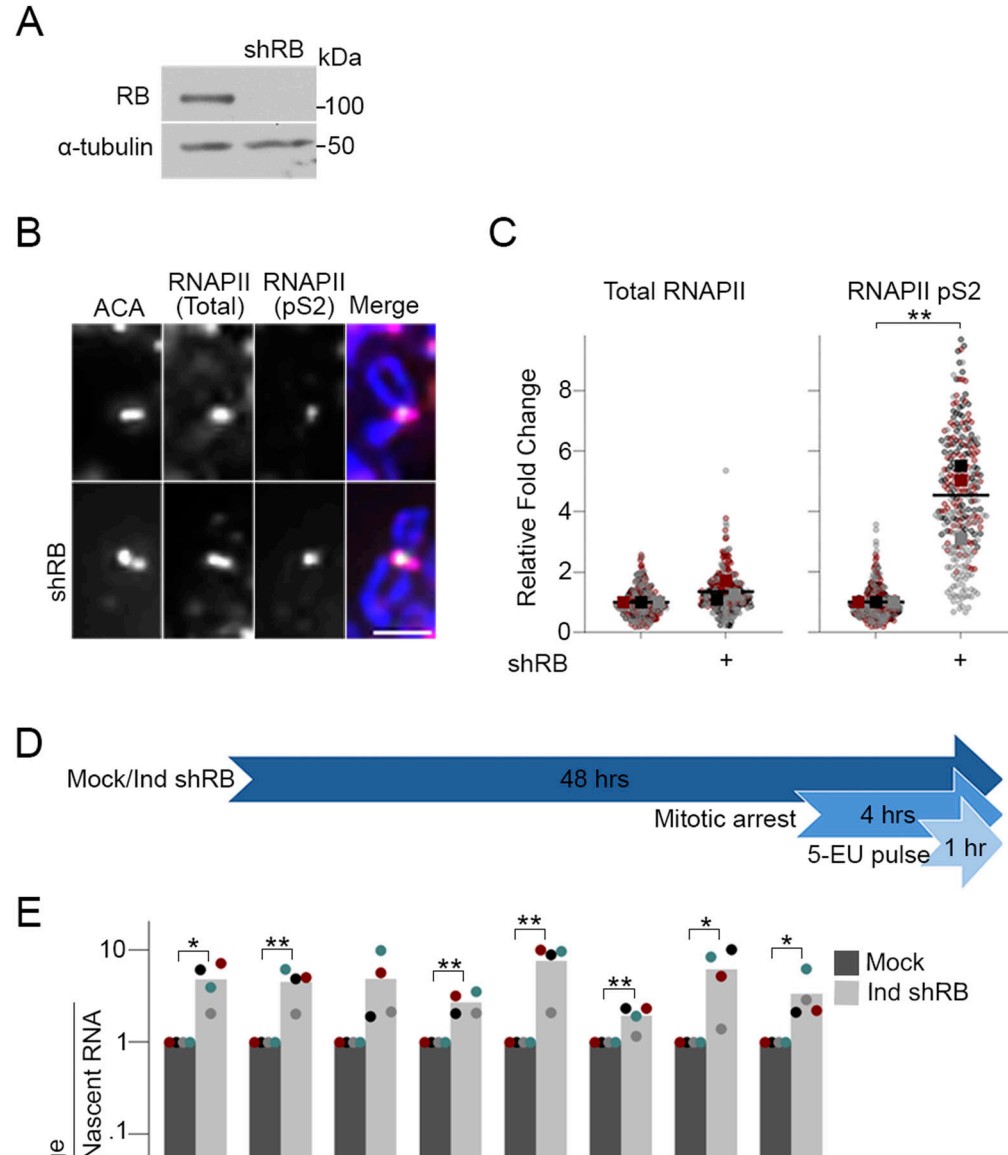

**Figure 1.  RB depletion increases RNAPII activity and transcription at the centromere.**
**(A)** RB depletion via shRNA (shRB) in hTERT-RPE-1 cells was verified by Western blot. **(B)** Representative images of mitotic spreads stained for total RNAPII (magenta), active RNAPII (pS2, green), ACA (red), and DNA (DAPI, blue). Scale bar = 2 $\mu$M. **(C)** Quantification of RNAPII and RNAPIIpS2 signal intensity across ACA-labeled centromeres. A minimum of 90 kinetochore pairs were measured (3/cell for 30 cells), for each of three biological replicates. **(D)** Schematic representation of protocol to label nascent mitotic RNA. **(E)** qRT-PCR analysis of nascent (EU-labeled RNAs) and total RNA in control and RB-depleted cells. Individual replicates are indicated by different colors, statistical analyses were performed between averages of biological replicates; *$P <$ 0.05, **$P <$ 0.01; ***$P <$ 0.001, ****$P <$ 0.001.

transcription is known to enhance R-loop-induced DNA damage (Crossley et al, 2019), we hypothesized that mitotic cells lacking RB may exhibit increased DNA damage at centromeres. To examine this possibility, hTERT-RPE-1 cells with and without induced expression of an RB-targeting hairpin (shRB) were arrested in mitosis for 4 h, fixed and stained for the canonical DNA damage marker γH2AX (Fig 2A), and number of damage foci per cell quantified. This analysis revealed that cells lacking RB exhibit an increase in the fraction of mitotic cells exhibiting DNA damage (classified as five or more γH2AX foci in an individual cell; Fig 2B). Complementary approaches to examine γH2AX foci on mitotic chromosome spreads from control and RB-depleted cells indicate that many of these damage foci are localized proximal to centromeres (Fig 2C and D).

Recent publications have demonstrated that increased transcription of alpha satellite DNA can occur in response to DNA damage (Yilmaz et al, 2021; Teng et al, 2024). Therefore, to discern whether increased R-loops and centromere transcription are a cause or consequence of DNA damage in our system, we examined cells following RNAPII inhibition. Importantly, the formation of R-loops at mitotic centromeres and the increase in mitotic DNA damage following RB depletion are both dependent on ongoing transcription during mitosis, as treatment of mitotic cells with the RNAPII inhibitor α-amanitin (administered concurrent with nocodazole-induced arrest) restored the proportion of cells exhibiting centromere R-loops and DNA damage to that seen in control cells (Figs S1 and 2A and 2B). Together these data suggest a relationship whereby aberrant centromere transcription and R-loop formation during mitosis have a causal relationship with DNA damage.

Whereas transcription-induced R-loops can lead to both single strand and double strand DNA breaks, γH2AX most efficiently labels double strand breaks (Rogakou et al, 1998; Sedelnikova et al, 2002). Therefore, to more comprehensively assess the extent to which RB-deficient cells acquire centromere and pericentromere nicks or breaks, we used a FISH approach, termed exoFISH. In this assay, non-denaturing conditions limit centromere-targeting FISH probes to hybridize only when single-stranded DNA sequences are revealed following in vitro digestion with exonuclease III (ExoIII). Since ExoIII gains access via nicks in the DNA backbone, preexisting ss or dsDNA breaks enable ExoIII-dependent digestion (Rogers & Weiss, 1980; Saayman et al, 2023b) and a corresponding increase in FISH probe hybridization. Using a probe that specifically targets the centromere-localized CENP-B binding sites present on most chromosomes, we examined mitotic chromosome spreads from both control (WT) and RB knockout hTERT-RPE cells (RB[KO]) (Nicolay et al, 2015) for evidence of centromere-proximal breaks (Fig 2E). The specificity and sensitivity of this assay in revealing the presence of pre-existing DNA breaks are supported by negative controls in which both WT and RB[KO] show low levels of centromere FISH probe hybridization in the absence of ExoIII treatment and positive controls in which WT and RB[KO] cells pretreated with the DNA nicking enzyme Nt.BsmAI exhibit comparable levels of centromere FISH probe hybridization following ExoIII treatment (Figs 2F and G and S2A and B). Using this assay, we find that centromeres in RB[KO] cells are more susceptible to exonuclease activity than control cells (Fig 2G). This increase in centromere FISH probe

accessibility is indicative of increased damage at or near mitotic centromeres.

R-loops recruit and activate the DNA damage response element ATR (Kabeche et al, 2018). During mitosis, ATR activity at centromeres stimulates AurB kinase to promote kinetochore-microtubule turnover (Cimini et al, 2006). In agreement with published reports (Kabeche et al, 2018), we find that ATR is recruited to mitotic centromeres in both control and RB-depleted cells (Fig 3A and B). However, consistent with observations that mitotic cells lacking RB exhibit high levels of transcription-dependent DNA damage (Fig 2), ATR activity at the mitotic centromere (as measured by the presence of the autophosphorylation mark pATR-T1989) is increased following RB depletion (Fig 3A and B). We additionally observe a comparable increase in recruitment of AurB kinase to mitotic centromeres (Fig S3A and B). Centromere transcription promotes CENPA deposition during G1 (Martins et al, 2020; Sidhwani & Straight, 2023), raising the possibility that increased CENPA deposition in RB-deficient cells may underlie the mitotic defects we observe. However, we do not observe changes in CENPA levels at the centromeres of RB-depleted mitotic cells (Fig S3C and D), indicating instead that changes in ATR and AurB localization that follow acute RB depletion (<2 cell cycles) are distinct and/or precede CENPA deposition. Importantly, the increase in ATR activity and AurB localization are transcription-dependent, as treatment with α-amanitin reduces centromere-localized pATR and AurB to levels seen in control cells (Figs 3A and B and S3). AurB kinase promotes kinetochore-microtubule turnover, such that increased AurB localization at the mitotic centromere enhances microtubule release and results in mitotic errors (Cimini et al, 2006; Knowlton et al, 2006; Muñoz-Barrera & Monje-Casas, 2014). Consistent with this, we and others find that RB-deficient cells exhibit a high rate of mitotic segregation errors (Fig 3C and D); (Hernando et al, 2004; Coschi et al, 2010; Manning et al, 2010). In agreement with a model whereby increased transcription/high levels of R-loops activate ATR and in turn enhance AurB activity, ATR inhibition does not impede R-loop formation (Fig S4), whereas anaphase defects in RB-deficient cells are similarly suppressed by either RNAPII (α-amanitin) or ATR (VE-821) inhibition (Fig 3C and D).

## Epigenetic silencing of centromeres suppresses changes that result from RB loss

Pericentromeres are enriched with heterochromatin-promoting histone modifications including H3K9me3 and H4K20me3 (Jeppesen et al, 1992; Rea et al, 2000; Peters et al, 2001; Rice et al, 2003; Schotta et al, 2004). The RB protein has been shown to physically and functionally interact with a number of chromatin modifiers, including the H4K20 methyltransferase Suv420h2 (Gonzalo et al, 2005; Isaac et al, 2006; Siddiqui et al, 2007), such that loss of RB leads to decreased Suv420h2 and H4K20me3 enrichment at pericentromeres and telomeres (Gonzalo et al, 2005; Isaac et al, 2006). Given the transcriptionally repressive role of H4K20me3 (Schotta et al, 2004; Sullivan & Karpen, 2004; Gonzalo et al, 2005; Kourmouli et al, 2005), the previously demonstrated role of RB in recruiting/sustaining Suv420h2 at centromeres during later stages

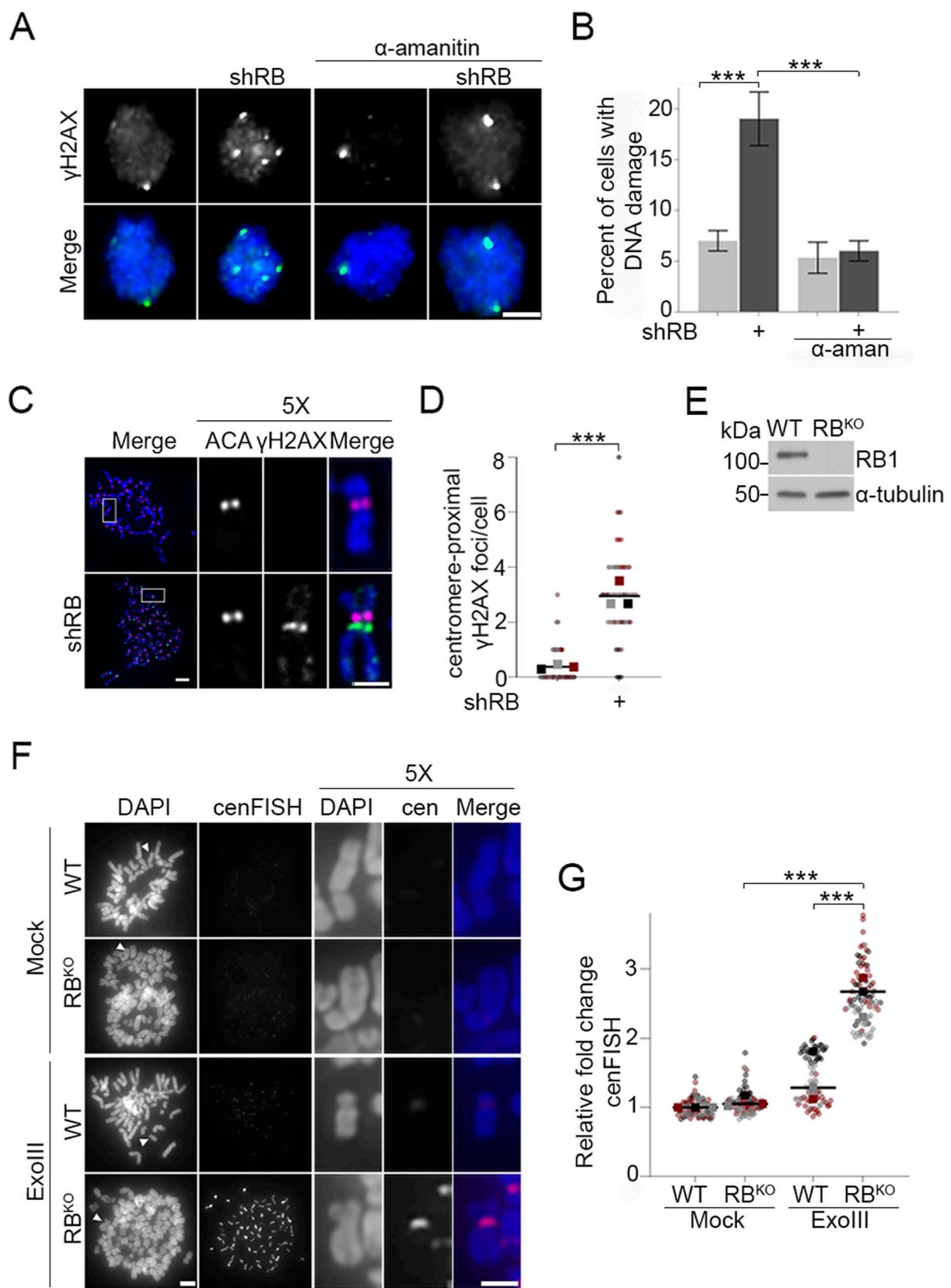

of the cell cycle (Gonzalo et al, 2005; Manning et al, 2010), and our observation that RB loss leads to transcriptional upregulation of mitotic centromeres, we sought to explore whether loss of Suv420h2 enrichment may underlie centromere deregulation when RB is lost or depleted. To this end, we used a previously established centromere-tethering system to anchor Suv420h2-GFP, via fusion to the DNA binding domain of centromere protein CENP-B, to centromeres (Herlihy et al, 2021). Using doxycycline-induced expression of a cen-Suv420h2-GFP fusion protein (Fig 4A), we first examined the impact of centromere-tethered Suv420-GFP (cenSuv) on mitotic centromere transcription in control and RB-depleted cells. As described above, we used two orthogonal approaches: immunofluorescence analysis of total and active (pS2) RNAPII at mitotic centromeres, and labeling and quantification of nascent centromere RNAs. These data revealed that mitotic centromere transcription, which is increased following RB depletion alone, is reduced by concurrent tethering of Suv420h2 to the centromere (Figs 4B and S5A–E).

Treatment with α-amanitin indicates that increased centromere damage, ATR activation, and increased centromere AurB localization that occur following RB loss are dependent on RNA polymerase II activity (Figs 2 and 3). However, α-amanitin treatment alone cannot discern between a role for centromere transcription and more general transcriptional deregulation that may occur throughout the genome when RB activity is lost. Therefore, to more explicitly test the role of centromere regulation on these attributes in RB-deficient cells, we assessed levels of mitotic DNA damage with immunofluorescence analysis of γH2AX foci and performed exoFISH, as described above, on control and siRB-depleted cells with and without expression of centromere-tethered Suv420h2-GFP. Upon induction of cen-Suv420h2-GFP the high level of γH2AX damage foci and ExoIII-dependent cenFISH signal seen in siRB depleted cells was reduced (Figs 4C and D and S6A–D).

We next used quantitative immunofluorescence to assess centromere levels of AurB kinase and anaphase defects following induced centromere tethering of Suv420h2-GFP in control and RB-depleted cells. Similar to RB depletion strategies described above (Fig S3), siRB-treated cells exhibit an increase in the staining intensity of AurB at centromeres (Fig 5A and B) and an increase in anaphase defects compared with control cells treated with an si-Scramble sequence (Fig 5C and D). Consistent with suppression of both centromere transcription and centromere-proximal DNA damage described above, centromere tethering of Suv420h2-GFP in siRB cells reduced both centromere-localization of AurB (Fig 5A and B) and anaphase defects to levels comparable to that observed in cells treated with a scrambled siRNA control alone (Fig 5C and D).

## Discussion

Here, we show that loss of the RB tumor suppressor permits increased RNA polymerase II activity at mitotic centromeres (Fig 1). The high temporal resolution afforded by first pulse labeling mitotic cells with 5-EU and then quantifying nascent RNA levels indicates that this feature of RB deficient cells corresponds with ongoing transcription at centromeres during mitosis (Figs 1 and 4). Our complementary approaches to either visualize the DNA damage marker γH2AX or alternatively to exploit ss and dsDNA breaks with exonuclease treatment reveal an increase in centromere-proximal DNA breaks following RB loss (Figs 2 and 4). We find that these breaks correspond with centromere-localized activation of ATR and AurB kinases (Figs 3, 5, and S2). Consistent with previously described roles for ATR and AurB in the regulation of mitotic chromosome segregation, RB deficient cells exhibit an increase in anaphase defects (Figs 3 and 5). Importantly, inhibition of bulk mitotic transcription during mitosis (RNA polymerase II inhibition with α-amanitin concurrent with mitotic arrest) or centromere-specific transcription (through centromere-tethering of the heterochromatin-promoting enzyme Suv420h2) limit mitotic DNA damage (Figs 2, 4, and S6), reduce ATR and AurB kinase activity at centromeres (Figs 3 and 5), and restore mitotic fidelity in RB-deficient cells (Figs 3 and 5). These data support a model whereby high levels of mitotic centromere transcription renders cells lacking RB sensitive to DNA breaks and ATR activation, leading to increased AurB kinase localization and activity at centromeres. Together, these changes promote chromosome segregation errors and contribute to chromosome instability (Fig 5E).

### RB-dependent regulation of heterochromatic boundaries at centromeres is critical for mitotic fidelity

Proteomics analysis has indicated that the RB tumor suppressor protein physically interacts with over 300 proteins, a significant portion of which are epigenetic modifying enzymes (Sanidas et al, 2019). Functional studies of the RB interactome suggest that RB may serve as a scaffold- moderating where in the genome and when in the cell cycle distinct epigenetic modifiers interact with chromatin (reviewed in Gonzalo and Blasco [2005]). Thus, by impacting the recruitment of enzymes that place transcriptionally repressive marks at centromeres (i.e., H3K27me3 by EZH2) (Blais et al, 2007; Ishak et al, 2016), H3K9me3 by Suv39 (Nielsen et al, 2001; Vandel et al, 2001), and H4K20me3 by Suv420h2 (Gonzalo et al, 2005; Manning et al, 2014), RB is poised to limit centromere and peri-centromere transcription. Consistent with work showing that disruption of RB interaction with EZH2 in mouse models deregulate

**Figure 2. RB loss leads to centromere proximal DNA breaks.**
**(A, B)** Representative images and quantification of γH2AX foci in mitotic hTERT-RPE-1 cells with or without induced shRNA-targeted depletion of RB (shRB) and/or treatment with α-amanitin (50 μg/ml) during mitosis. Scale bar = 5 μm, error bars represent SD between biological replicates. **(C, D)** Representative images and quantification of centromere-proximal γH2AX signal (green) in metaphase spreads co-stained for ACA (magenta) and DNA (DAPI, blue). A minimum of 90 kinetochore pairs were measured (3/cell for 30 cells), for each of three biological replicates. Scale bar = 5 μm in full spread, 2 μm in enlargement. **(E)** Western blot validation of RB loss in CRISPR-RB knockout hTERT-RPE cells (RB^KO). **(F, G)** Representative images and quantification of cenFISH probe labeling in WT and RB^KO cells following treatment (or not) with Exonuclease III. A minimum of 180 kinetochore pairs were measured (6/cell for 30 cells), for each of three biological replicates. In panels (C, F) white arrowheads or boxes indicate chromosomes represented in enlargements, scale bars are 5 μm in full spread, 2 μm in inset. Individual replicates are indicated by different colors, statistical analyses were performed between averages of biological replicates; **P < 0.01, ***P < 0.001.

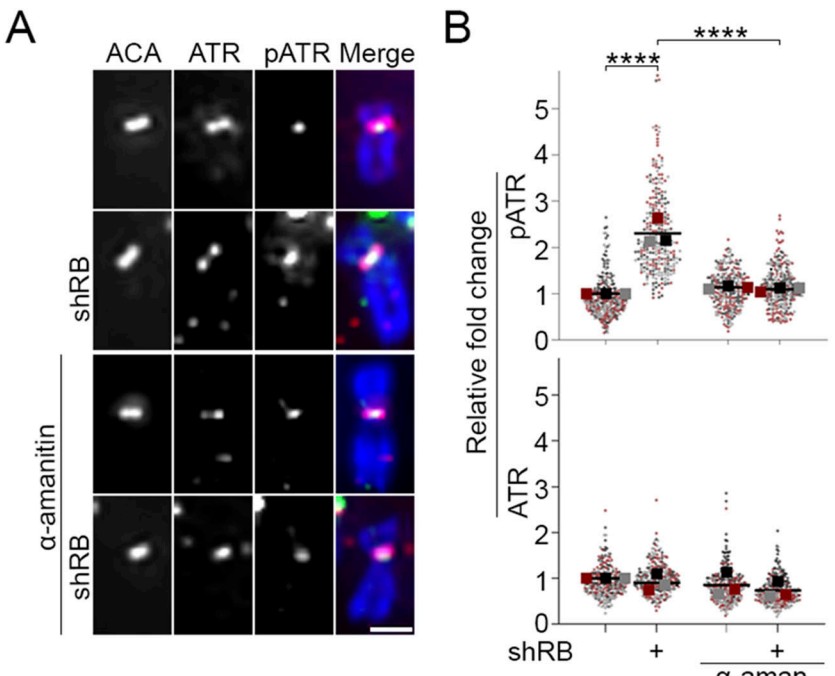

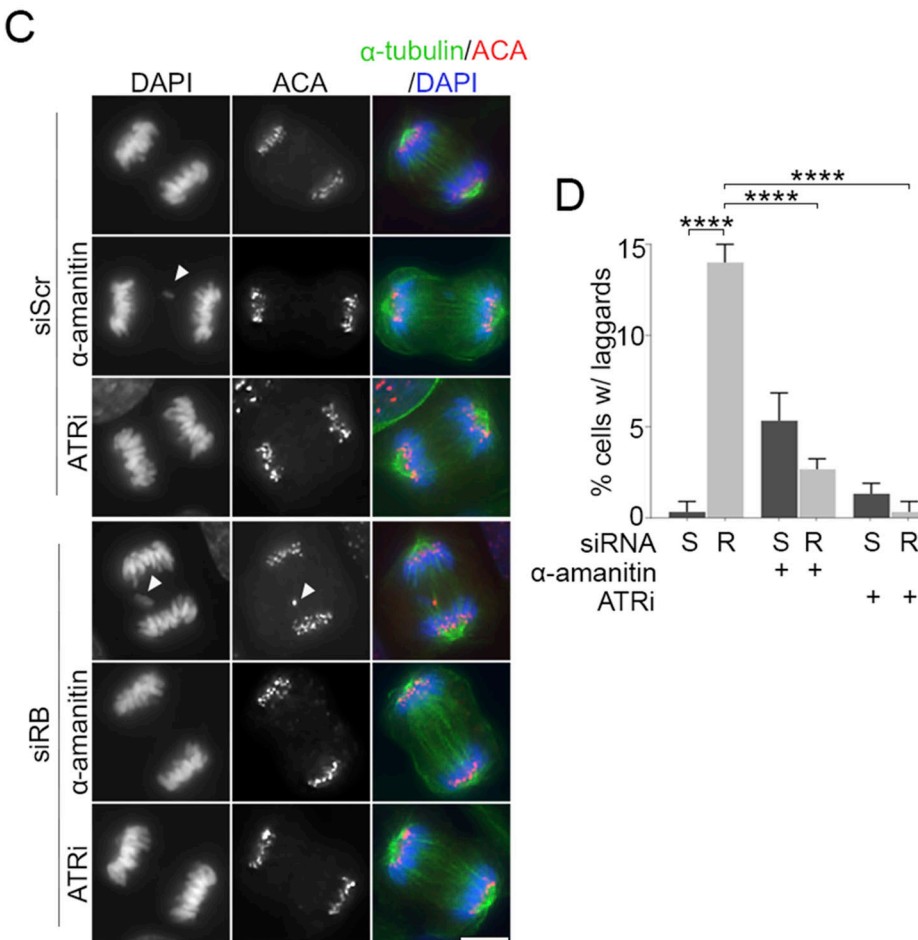

**Figure 3. Transcription-dependent ATR activation compromises mitotic fidelity following loss of RB.**

**(A, B)** Representative images and quantification of total (magenta) and phosphorylated (T1989; green) ATR at mitotic ACA-labeled (red) centromeres in control and RB-depleted hTERT-RPE-1 cells. Cells were untreated or treated with the RNA polymerase II inhibitor α-amanitin (50 μg/ml). A minimum of 90 kinetochore pairs were measured (3/cell for 30 cells), for each of three biological replicates. Scale bar = 2 μm. **(C, D)** Representative images and quantification of anaphase defects in hTERT-RPE-1 cells treated with either a non-targeting control (S) or RB-specific (R) siRNA and subsequently treated with ATR inhibitor (VE-821, 10 μM) for 1 h, or α-amanitin (50 μg/ml) for 4 h. A minimum of 50 cells were scored per condition for each of three biological replicates. White arrowheads indicate lagging chromosomes, scale bar = 5 μm. Error bars represent SD between biological replicates, and statistical analyses were performed between averages of biological replicates, ****$P < 0.0001$.

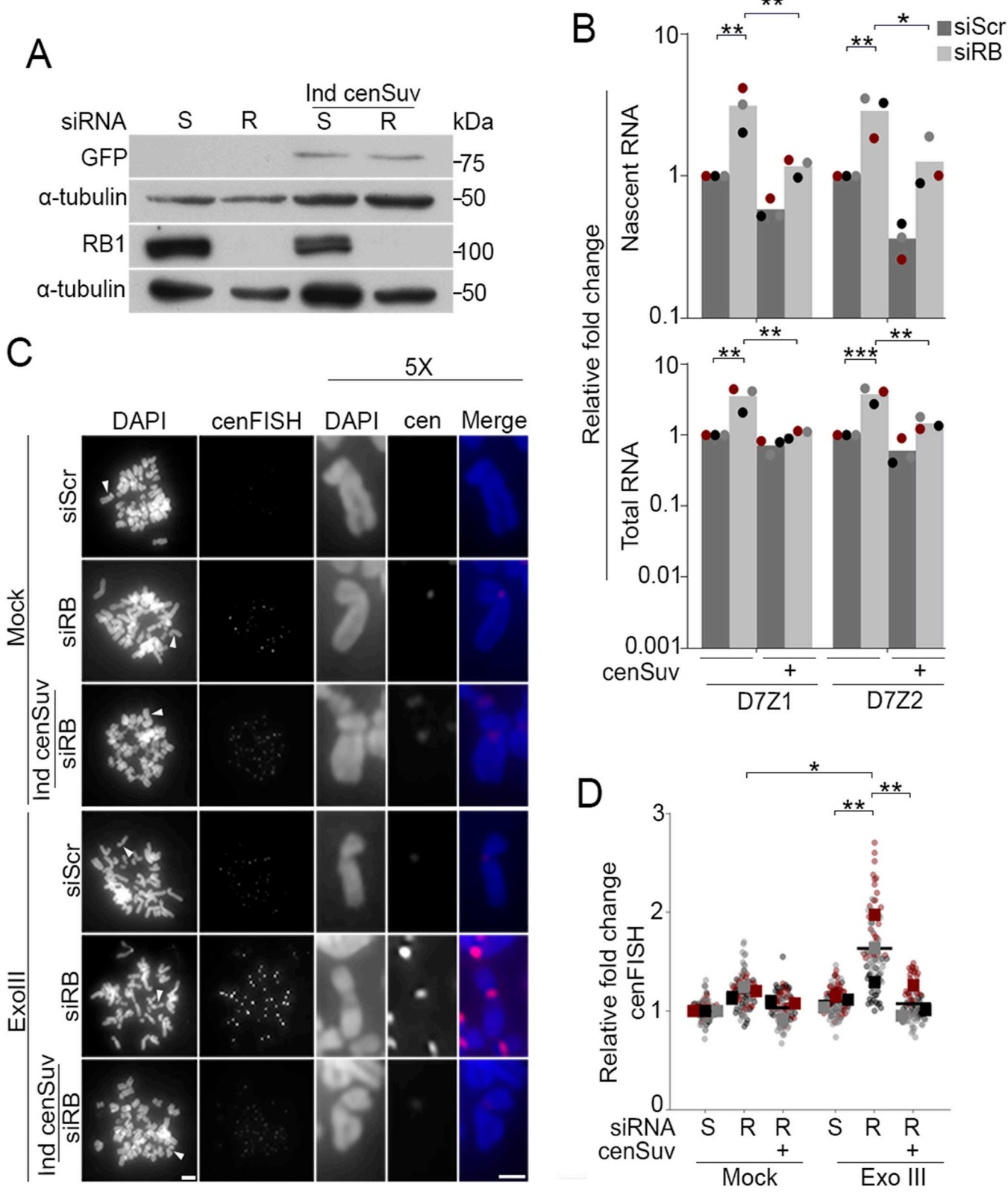

**Figure 4. Decreasing mitotic centromere transcription reduces centromere breaks.**
**(A)** Western blot analysis confirmation of RB knockdown and Suv420h2 overexpression in hTERT-RPE + inducible cen-Suv420h2-GFP cells treated with either a non-targeting control (S) or RB specific (R) siRNA, with or without induction of cen-Suv420h2-GFP expression. **(B)** qRT-PCR analysis of nascent (5-EU-labeled RNA) and total RNA transcribed from a representative centromere (D7Z1) and pericentromere (D7D2) in mitotic cells. **(C, D)** Representative images and quantification of cenFISH signal in control and RB-depleted cells with or without Exonuclease III treatment. A minimum of 180 kinetochore pairs were measured (6/cell for 30 cells), for each of three

transcriptional repression (Ishak et al, 2016), we demonstrate here that loss of RB permits high levels of transcriptional activity at the centromeres in mitotic human cells and leads to defects in chromosome segregation during cell division. Furthermore, rescue experiments that exploit mitotic specific (via short term α-amanitin treatment) or centromere-localized (via centromere-tethered Suv420h2/centromere-specific H4K20me3 enrichment) transcriptional repression to limit segregation defects indicate that mitotic errors following RB loss result from deregulation of centromere transcripts late in the cell cycle, and may not otherwise be dependent on global changes in protein expression that occur from loss of RB-dependent repression of E2F transcription factors earlier in the cell cycle.

### Balancing centromere transcription for genome and chromosome stability

Prior studies have demonstrated that inhibition of RNAPII and the corresponding decrease in centromere transcription promote mitotic errors (Rošić et al, 2014; McNulty et al, 2017; Chen et al, 2021). These studies implicate centromere transcription in the establishment of open chromatin that is conducive for CENP-A deposition in preparation for the subsequent cell cycle (Bobkov et al, 2018), and the transcripts themselves in recruiting and tethering critical kinetochore components (Ferri et al, 2009; Jambhekar et al, 2014; Quénet & Dalal, 2014; Liu et al, 2015; Blower, 2016; McNulty et al, 2017). In the absence of centromere-derived transcripts, the AurB kinase-containing chromosomal passenger complex (CPC) is not recruited to the kinetochore (Jambhekar et al, 2014; Blower, 2016). AurB kinase functions to destabilize kinetochore microtubules, a critical step in releasing improper attachments that form early in mitosis. In the absence of AurB activity, kinetochore-microtubule attachments are hyper stabilized and improper attachments persist, ultimately delaying biorientation and leading to segregation errors (Hauf et al, 2003; Abe et al, 2016; Huang et al, 2018; Broad et al, 2020).

In contrast to work showing that mitotic fidelity is sensitive to reduction of centromere transcription, de-repression and/or increased expression of repetitive sequences derived from or near centromeres, including LINE elements Satellite repeats, has been observed in various cancer contexts and is strongly correlated with poor patient prognosis (Ting et al, 2011; Zhu et al, 2011, 2018; McNulty et al, 2017). While current studies have not yet discerned whether high levels of centromere transcription may be a driving force in tumorigenesis or merely a passenger that indicates widespread deregulation of transcriptional repression, recent molecular studies highlight several models whereby centromere transcription may directly impact genome stability and underlie cancer susceptibility (reviewed in Petermann et al [2022]). First, the process of transcription results in the generation of a DNA:RNA hybrid and displaced ssDNA structure known as an R-loop. These structures are sensitive to both single strand and double strand DNA breaks and during S phase additionally pose an obstacle to

replication that can promote replication stress and further DNA damage. To mitigate this risk, cells actively limit aberrant or excessive R-loop accumulation through the activity of DNA:RNA hybrid-specific endoribonuclease activity (Lockhart et al, 2019). Second, the presence of R-loops activates the DNA damage response. During mitosis, this includes the recruitment of the ATR kinase to centromeres. ATR activity leads to local activation of AurB kinase (via an ATR-Chk1 axis) (Kabeche et al, 2018). As described above, AurB kinase functions to destabilize kinetochore microtubule attachments (Muñoz-Barrera & Monje-Casas, 2014) which can in turn compromise mitotic fidelity (Muñoz-Barrera & Monje-Casas, 2014; González-Loyola et al, 2015; Crowley et al, 2022). Activated oncogenes have been described to increase transcription and R-loop formation, leading to replication stress and genomic instability (reviewed in Petermann et al [2022]). Our work additionally implicates the RB tumor suppressor, which is commonly lost or functionally inactivated across a broad range of cancer contexts (Burkhart & Sage, 2008), as a regulator of mitotic centromere transcription. We propose that, through recruitment of key epigenetic modifying enzymes (such as Suv420h2) during later stages of the cell cycle (Gonzalo et al, 2005; Isaac et al, 2006), RB functions to establish and/or maintain a heterochromatic boundary at centromeres that in turn limits centromere transcription and AurB activity to promote mitotic fidelity. A number of groups have described that loss of RB compromises mitotic fidelity (Manning & Dyson, 2011). Data presented here are consistent with a model whereby, in the absence of RB, repressive marks near centromeres are reduced permitting excessive transcriptional activity. Aberrant R-loop formation and the corresponding recruitment of ATR kinase may then collaborate with transcript-dependent recruitment of AurB to destabilize kinetochore microtubules, concurrently leading to both centromere damage and whole chromosome segregation errors.

# Materials and Methods

### Cell culture

hTERT immortalized RPE-1 RB[KO] (gift from the Dyson lab, Massachusetts General Hospital Cancer Center), RPE shRB (Zamalloa et al, 2023), and RPE cen-Suv420h2 GFP (Herlihy et al, 2021) cells were grown in DMEM (Gibco) supplemented with 10% FBS (Sigma-Aldrich) and 1% penicillin/streptomycin (Gibco). All cell lines were maintained at 37°C and 5% $CO_2$. High resolution immunofluorescence imaging with DNA stain (DAPI, Thermo Fisher Scientific) was used to monitor and confirm cell lines were free of *Mycoplasma* contamination. Over expression of cen-targeted Suv420h2 GFP was achieved by treatment with 2 μg/ml doxycycline for 16 h. Inhibition of ataxia telangiectasia mutated and Rad3-related kinase (ATR) was achieved through treatment with VE-821 at a final concentration of 10 μM (Sigma-Aldrich) for 1 h. The inhibition of RNA

biological replicates. Scale bars are 5 μm for whole spread panels and 2 μm for individual chromosome enlargement. Individual replicates are indicated by different colors, statistical analyses were performed between averages of biological replicates; *P < 0.05, **P < 0.01; ***P < 0.001.

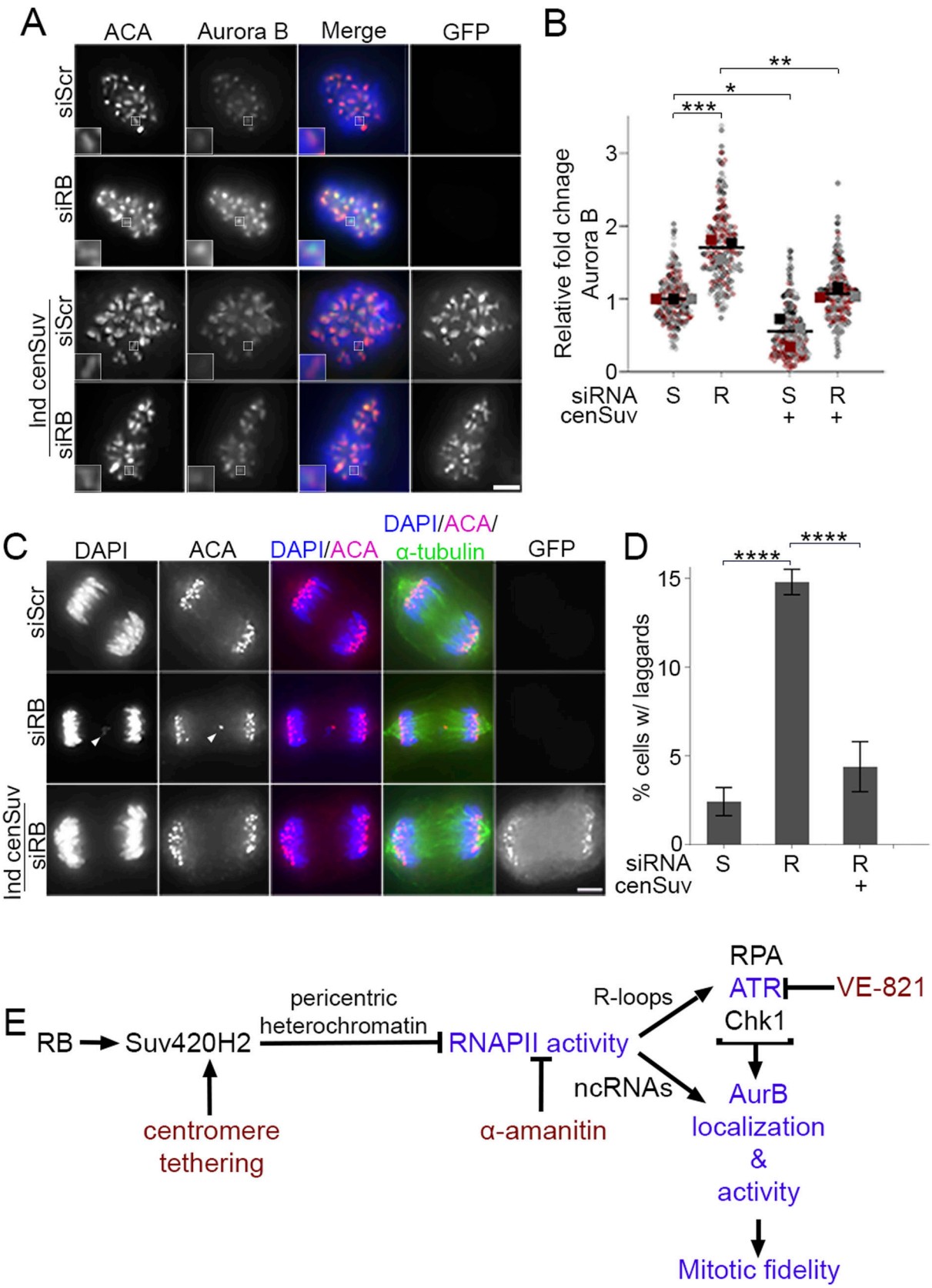

polymerase II transcription was completed using a final concentration of 50 µg/ml α-amanitin (Santa Cruz) for 4 h. Depletion of RB1 was achieved through transient transfection with 50 nM pool siRNAs (a pool of 4 siRNAs to RB1; siRNA-SMARTpool, Horzion Discovery) using RNAiMax transfection reagent (Thermo Fisher Scientific) according to the manufacturer's instructions. Transfection with a SMARTpool of four non-targeting siRNA sequences (#AM4636; Ambicon) was used as a negative control for all depletions (Table S1). Alternatively, cells were infected with a lentiviral construct containing a doxycycline inducible shRNA hairpin (tet-pLK0-Puro, #21915; Addgene) for the targeted depletion of RB. Stable hairpin-expressing cells were selected with Puromycin for 10 d. Induced depletion was achieved by the addition of 2 µg/ml doxycycline for a minimum of 48 h.

## Immunoblotting

Cell extracts were prepared using 2x Laemmli buffer (Bio-Rad) with β-Mercaptoethanol (Sigma-Aldrich). Protein concentrations were normalized to total cell number and samples run on an SDS–PAGE gel. Proteins were transferred to PVDF membrane (Millipore) and blocked in 1xTBST supplemented with 5% milk. Antibodies were diluted 1:1,000 in 1xTBST/5% milk: DM1Aα (α-tubulin, Santa Cruz), RB1 (4H1, 9009; Cell Signaling), GFP (D5.1; Cell Signaling) and incubated at 4°C. Membranes were washed in 1xTBST, incubated for in corresponding HRP-conjugated secondary antibody (GE Healthcare), and developed using ProSignal Pico (Prometheus).

## Immunofluorescence and metaphase spreads

Cultured cells were grown on coverslips, fixed, and stained for AurB (611083; BD Biosciences) and ACA (15-243; Antibodies Inc) as previously described in Kleyman et al (2014) or for CENPA (Enzo ADI-KAM-CC006-E) as described in Crowley et al (2022). To visualize anaphase defects and monitor DNA damage in nocodazole arrested prometaphase cells were fixed with 4% PFA for 20 min, extracted with 0.2% TritionX-100 in PBS for 10 min, and blocked with TBS + 1% BSA. Primary antibodies, ACA (1:500, 15-243; Antibodies Inc), DM1A (1:1,000 sc-32293; Santa Cruz), or γH2AX Ser139 (1:1,000, 2577; Cell Signaling) were diluted in TBS +1% BSA. Secondary antibodies were diluted in 1% BSA + 0.2 mg/ml DAPI and coverslips were mounted onto slides using Prolong Antifade Gold (Molecular Probes). Mitotic cells were collected via shake-off after treatment with 100 ng/ml nocodazole (Selleckchem) for 3 h and metaphase spreads prepared as in Keohane et al (1996). Primary and secondary antibodies were diluted in KCM + 1% BSA. For staining of total (2B5, 1:

200 GTX70109; GeneTex) or phospho ATR (pT1989, 1:200 128145; GeneTex), buffers were supplemented with a final concentration of 100 nM Calyculin A (Millipore Sigma) (Kabeche et al, 2018). For total (F-12, 1:200 sc-55492; Santa Cruz) or active RNAPII (pS2, 1:200 ab5095; Abcam) staining, buffers were supplemented with 100 nM Calyculin A (Millipore Sigma), 40 U/µl of RNasin (Promega) and kept on ice (Chan et al, 2012; Perea-Resa et al, 2020). Cells were then post fixed with 4% PFA for 10 min before counter staining with TBS + 5% BSA + 0.2 mg/ml DAPI for 30 min. Coverslips were mounted onto slides using Prolong Antifade Gold (Molecular Probes). Fixed cell images were captured using a Zyla sCMOS camera mounted on a Nikon Ti-E microscope, with a 60X Plan Apo oil immersion objective 1.4NA and 0.3 µm z-stacks. To assess centromeric protein levels, NIS-elements Advanced Research software was used to perform line scans in a single focal plane through individual ACA-stained kinetochore pairs where the area under the curve indicates the region of centromere/kinetochore-localized staining. γH2AX levels were assessed by counting the number of foci per cell. A cell was considered damaged if it had more than 5 foci per cell. For intensity measurements, a minimum of three kinetochore pairs per 30 cells, per condition (90 kinetochore pairs/condition) were measured in each of three biological replicates. Anaphase defects were assessed in a minimum of 50 cells per condition for each of three biological replicates.

## exoFISH

exoFISH was performed as described in Saayman et al (2023a) with the following modifications. Cells were first prepared and spread onto coverslips as in Ganem et al (2009) Slides were then dried overnight at RT in the dark. The next day the slides were rehydrated in 1X PBS, treated with 0.5 mg/ml RNaseA (NEB) for 10 min at 37°C in a humid chamber, washed with 1X PBS, then treated or not with 1 unit of Nt.BsmAI (NEB) for 2 h at 37°C in a humid chamber. Slides were washed with 1X and treated or not with 200 mU/µl ExoIII diluted in 1X buffer supplied by the manufacturer for 1 h then dehydrated overnight. To visualize breaks at the centromere, slides were incubated in 0.5 µM CENPB-binding site specific cenFISH probe (PNA Bio) for 3 h at RT. Following incubation in hybridization wash buffers and subsequent dehydration, cover glass was then mounted onto slides using Prolong Antifade Gold. To quantify cenFISH signal, intensity values were measured within six 12 × 12 pixel boxes placed at centromeres and summed per cell for 30 cells per condition for each of three biological replicates. Fold change for each condition was calculated by normalizing to the RPE-1 without exoIII condition for that replicate.

**Figure 5. Suppression of transcription at the centromere limits Aurora B localization and reduces lagging chromosomes.**
**(A, B)** Representative images and quantification of centromere-localized Aurora B (green) in hTERT-RPE + cen-Suv420h2-GFP cells treated with either a non-targeting control (S) or RB specific (R) siRNA, with or without induction of centromere tethered Suv420h2 and co-stained for ACA (red), GFP (white), and DNA (DAPI, blue). Insets are of individual kinetochore pairs at 3X magnification. A minimum of 90 kinetochore pairs were measured (3/cell for 30 cells), for each of three biological replicates. **(C, D)** Representative images and quantification of anaphase defects in cells stained for ACA (magenta), tubulin (green), and DNA (DAPI, blue). A minimum of 50 cells were scored per condition for each of three biological replicates. Scale bars are 5 µm. Error bars represent SD between biological replicates. Individual replicates are indicated by different colors, statistical analyses were performed between averages of biological replicates; *P < 0.05, **P < 0.01; ***P < 0.001, ****P < 0.0001. **(E)** Model proposing how RB-dependent regulation of centromere transcription promotes mitotic fidelity. Dark blue text represents experimental readouts described in this study; red font represents experimental manipulations used to test/establish relationships.

### Quantification of nascent and total RNA levels

For quantification of nascent and total RNA levels, mitotic cells were first isolated via mitotic shake off following incubation in 100 ng/ml nocodazole final concentration for 4 h and subsequent treatment with 0.25 µM 5-ethynyl uridine (5-EU; Vector Labs). Cells were processed according to the procedure outlined by the Click-iT Nascent RNA Capture Kit (Invitrogen). 2–5 µg of RNA was removed before the remainder of the RNA undergoing the Click-iT reaction to be used for total RNA quantification and confirmation of RB knockdown. Total RNA was treated with DNaseI (NEB) before cDNA was synthesized from 1 µg of total RNA using SuperScript IV Reverse Transcriptase (Invitrogen) according to the manufacturer's instructions. cDNA synthesis for nascent RNA was performed according to the manufacturer's instructions. Gene expression for centromeric and peri-centromeric transcripts (Table S1) was determined using the ΔΔ cycle threshold method and normalized to GAPDH.

### DRIP analysis

Cells were isolated via mitotic shake off following incubation in 100 ng/ml nocodazole for 4 h. DNA:RNA hybrid immunoprecipitation was performed on 8 µg of purified chromatin using 10 µg S9.6 antibody (ENH001; Kerafast) or mouse IgG (sc-516176; Santa Cruz) and Protein A Dynabeads, as previously described (Sanz & Cehdin, 2019). To verify specificity of DNA:RNA hybrid isolation, parallel IPs were performed with and without pretreatment with 40U RNasH1 to degrade DNA:RNA hybrids. Centromeric and peri-centromeric sequences were analyzed by qRT-PCR (primers in Table S1) using the ΔΔ cycle threshold method and represented as fold change % input relative to the IgG control.

Experimental data were analyzed with a *t* test or one-way ANOVA where appropriate. Individual measurements from experiments where multiple measurements were made per replicate are represented as SuperPlots, with individual replicates color-coded. Per-replicate averages and SD between biological replicates are superimposed. All error bars represent SD between biological replicates and statistically significant differences are labeled with *$P < 0.05$, **$P < 0.01$, ***$P < 0.001$, and ****$P < 0.0001$.

## Data Availability

Raw data underlying this work are available from the corresponding author upon reasonable request.

## Supplementary Information

## Acknowledgements

The authors thank Nicole Hermance and Rachel Flynn for technical support and critical evaluation of the manuscript. This work was supported by the American Cancer Society (RSG-21-066-01-CCG) and the National Science Foundation (2143869) awards to AL Manning, Dr Helen G Vassallo Distinguished Presidential Professor.

### Author Contributions

EA Crowley: conceptualization, data curation, formal analysis, validation, investigation, visualization, methodology, and writing – original draft, review, and editing.
AL Manning: conceptualization, data curation, supervision, funding acquisition, methodology, project administration, and writing – original draft, review, and editing.

### Conflict of Interest Statement

The authors declare that they have no conflict of interest.

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
