## [Reviewer comments · Life Science Alliance]

RB dependent transcriptional regulation at mitotic centromeres preserves genome stability

Elizabeth Crowley and Amity Manning

DOI: <https://doi.org/10.26508/lsa.202503433>

Corresponding author(s): Amity Manning, Worcester Polytechnic Institute

Review Timeline:

Submission Date:	2025-06-25
Editorial Decision:	2025-07-31
Revision Received:	2025-10-31
Editorial Decision:	2025-12-02
Revision Received:	2025-12-10
Accepted:	2025-12-12

Scientific Editor: Sarita Hebbar

Transaction Report:

July 31, 2025

Re: Life Science Alliance manuscript #LSA-2025-03433-T

Dr. Amity L Manning
Worcester Polytechnic Institute
Biology and Biotechnology
60 Prescott St
Worcester, MA 01605

Dear Dr. Manning,

Thank you for submitting your manuscript entitled "RB dependent transcriptional regulation at mitotic centromeres preserves genome stability" to Life Science Alliance. The manuscript was assessed by three expert reviewers, whose comments are appended to this letter.

As you will note, the reviewers find this work of potential significance. However they have raised some concerns that need to be addressed before publication.

Reviewers 1 and 3 noted that corroboration for existing data, describing the effect of Rb on centromeric transcription, will be required. Their suggestions include immunofluorescence experiments (RNAPII pS2 and Rb protein localisation at centromeres). We agree that you should provide the data on RNAPII pS2 localisation. We leave it to your decision to either provide Rb localisation data or discuss the possibility that the effects on centromeric transcription maybe an indirect consequence of Rb depletion.

We further concur with the requests made by Reviewers 1 and 3 that you should provide data from orthogonal approaches to study R-loop formation and changes following RB depletion in cells. You can use a method of your choice from the suggested options.

Finally, we agree that the points identified by all the reviewers for additions to the discussion section are necessary.

In line with their overall assessment, we invite you to submit a revised manuscript addressing the reviewers' comments. When submitting the revision, please include a letter addressing the reviewers' comments point by point. While a rebuttal must respond to all points in some form, additional experiments to resolve these points, other than indicated above, will not be required.

Thank you for this interesting contribution to Life Science Alliance. We are looking forward to receiving your revised manuscript.

Sincerely,

Sarita Hebbar, PhD
Scientific Editor
Life Science Alliance
<http://www.lsjournal.org>

B. MANUSCRIPT ORGANIZATION AND FORMATTING:

Reviewer #1 (Comments to the Authors (Required)):

In this manuscript Crowley et al. investigate the role of centromeric transcription in maintaining mitotic fidelity and its perturbation following the loss of the tumor suppressor RB. Building on a growing body of literature that highlights the necessity of RNAPII-mediated transcription at centromeres for proper CENP-A deposition and kinetochore assembly, the authors demonstrate that RB depletion leads to elevated transcription at both centromeric and pericentromeric regions. This transcriptional upregulation correlates with increased recruitment of pATR and Aurora B, factors associated with replication stress and mitotic surveillance. Broadly, we found the premise of this work very interesting, and in line with growing evidence in the field linking cancer pathways to regulation of CENP-A deposition. Congratulations on a nicely executed study.

Three broad points the authors might wish to consider:

- 1) This study presents compelling evidence of centromere dysfunction, particularly relevant in cancer cells where RB is frequently mutated or functionally inactivated. However- it will be beneficial to look at actual cancer cells where Rb is mutated to determine the pathological significance. We accept that this may be beyond the scope of the current study, but maybe discuss this a bit further?
- 2) the authors propose that RB loss promotes R-loop formation at these repetitive loci, which in turn drives aberrant transcription and genome instability. This is where the manuscript could use a bit of more direct evidence for R-loop formation at centromeres or pericentromeres in RB-deficient cells. Incorporation of R-loop mapping (e.g., DRIP-seq) would significantly strengthen this model significantly and hugely elevate the importance and impact of this work.
- 3) the authors also address the pathological consequences of dysregulated centromere transcription. The authors highlight that excessive expression of centromeric and pericentromeric repeats, often observed in cancer, leads to the formation of R-loops-structures that trigger replication stress, activate ATR kinase, and ultimately enhance Aurora B activity. While Aurora B is necessary for error correction, its hyperactivation, particularly in mitosis, can paradoxically promote chromosome segregation errors and genome instability. Again, this is a bit of a weak link, as the mechanistic pathway linking R-loop formation to Aurora B activation, however, remains to be clearly defined. Although the study centers on centromere integrity, the authors do not report on key inner kinetochore components such as CENP-A or CENP-C, whose misregulation could provide further insight into mitotic errors observed in RB-deficient cells especially in the context of increased transcription. We encourage the authors to consider adding such data that we think would make this work a whole lot more insightful.

Other experimental or technical points to consider:

1. In Figure 1 the authors show primarily by immunofluorescence that RNAP II pS2 is upregulated at centromeres upon Rb depletion. A better way to quantitatively assay increased RNAPII pS2 upon depletion of RB is to perform ChIP qPCR at the centromeric DNA for both RNA pol II and RNAPII pS2. This can be incorporated in the manuscript.
2. R-DNA loops can form also by RNA pol II pausing (Xu, C., Li, C., Chen, J. et al. Nature 621, 610-619 (2023). <https://doi.org/10.1038/s41586-023-06515-5>). The authors should delineate in this study that whether there is RNAP II pausing or not at the centromeres
3. In Figure 1E the authors measure the nascent RNA levels in absence of RB. The authors should also determine the

expression of LINEs and SINEs elements at the centromeres or pericentromeres.

4. The authors although later in the manuscript comment on the link between RB and recruitment of certain epigenetic enzymes, however it will be better to show the link at the endogenous level. What happens to H4K20me3, H3K9me3 at the centromeres when cells are depleted of the RB protein?

5. In figure 2C the labelling of H2AX and ACA is not proper. The authors should be careful in labelling the figures as that might lead to erroneous interpretation of the data.

6. In Figure 2E the authors employed a fluorescent in situ hybridization (FISH) approach, exoFISH. To confirm the formation of R-DNA loops the authors should use other direct methods like DRIP seq or using GFP-catalytically dead RNase H1 for visualization at the centromeres in siRB cells.

7. Also, another way to relieve R-DNA loop formation is treating cells with Topoisomerase inhibitors. The authors should also assay the transcription at the centromeres upon these inhibitors in the Rb deficient cells to confirm that it is indeed through the R-DNA hybrids

8. The authors also do not show how the enhanced transcription at the centromere or pericentromeres affect CENP-A/C deposition as they are both intricately co regulated.

9. In figure 3A and 3B the authors show that in absence of RB the pATR is high, when ATRi the laggards are decreased. However, the authors do not comment on the R DNA loop formation upon ATRi, does it decrease also?

10. RB (retinoblastoma protein) and E2F pathway plays a crucial role in regulating the cell cycle, particularly the transition from G1 phase to S phase. Do the authors here characterize the cell cycle of the Rb deficient cells? Also for the mitotic defects are laggards the only mitotic defects they observe in the Rb deficient cells?

11. In figure 4 and 5 the authors show that tethering the Suv420H2-GFP at the centromeres reduced the ExoIII-dependent cenFISH signal and the nascent RNA transcription. Subsequently a reduction of AurKB at the centromeres. A direct interaction of Suv420 and RB at centromeres are not shown. Also, in siRB cells does the Suv420 endogenous levels change at the centromeres?

12. RB is often mutated in cancer cells, validating this model in any cancer cells harboring RB mutations is a good model to validate the findings of this study.

Reviewer #2 (Comments to the Authors (Required)):

RB deficiency is associated with chromosomal instability. Previous studies, including work by the authors, demonstrated that this instability can be partly attributed to defects in chromosome segregation. While these defects were linked to deregulation of heterochromatic marks and altered cohesin binding at pericentromeric regions, the precise mechanism by which RB safeguards chromosome segregation fidelity remained unclear.

In this study, the authors conducted a detailed analysis revealing that RB epigenetically regulates transcriptional activity at centromeres, thereby modulating ATR and AurB kinase activity to prevent mitotic errors. Specifically, they demonstrate that:

- RB depletion increases RNAPoIII activity at mitotic centromeres.
- Mitotic RB-deficient cells show a marked increase in newly synthesized centromeric and pericentromeric transcripts.
- RB-deficient cells exhibit a higher proportion of mitotic cells with DNA damage.
- Inhibition of RNAPoIII with α -amanitin in RB-deficient cells restores DNA damage levels to those seen in control cells.
- Centromeres in RB-deficient cells are more sensitive to exonuclease digestion.
- ATR activity and recruitment of AurB kinase at mitotic centromeres are elevated upon RB loss.
- ATR activation and AurB localization are dependent on centromeric transcription.
- Anaphase defects in RB-deficient cells are suppressed by inhibition of either RNAPoIII or ATR.
- Tethering Suv420H2, an RB-bound histone methyltransferase, to centromeres reduces RB loss-induced transcriptional upregulation at mitotic centromeres.
- Suv420H2 tethering also decreases centromeric DNA damage in RB-deficient mitotic cells.
- Finally, Suv420H2 recruitment to centromeres reduces AurB localization and anaphase defects in RB-deficient cells.

The proposed mechanism reveals a previously unrecognized role of RB during mitosis and provides important insights into the consequences of RB loss in tumors, particularly those marked by genomic instability. A key observation, which is appropriately emphasized in the discussion, is that the rescue strategies used to mitigate segregation defects in RB-null cells specifically target mitosis (e.g., RNAPoIII inhibition in nocodazole-arrested cells) and centromere-localized epigenetic regulation (via cen-Suv420H2 expression). These findings suggest that mitotic errors following RB loss arise primarily from dysregulated centromeric transcription late in the cell cycle, rather than from canonical RB-mediated repression of E2F target genes. This supports a novel, non-canonical role for RB in safeguarding mitotic fidelity.

Although the timing of Suv420H2 recruitment to centromeres remains undefined, the authors could consider discussing whether this function reflects a mitosis-specific activity of RB or could involve earlier, interphase-mediated recruitment mechanisms.

Overall, the manuscript is well written, and the authors' conclusions are well supported by the data. I co-reviewed this manuscript with a postdoctoral fellow in my lab, and we both strongly recommend it for publication.

Minor comment:

The catalog numbers for the commercially available primary antibodies used in this study are missing and should be included for reproducibility.

Reviewer #3 (Comments to the Authors (Required)):

This study tried to build up a story in which Rb depletion results in mitotic defects via centromeric transcription upregulation. Specifically, the authors claimed that Rb depletion upregulates centromeric transcription during mitosis, which increases the amount of R-loops that lead to DNA damage at centromeres. DNA damage then activates ATR at centromeres, which recruits more Aurora B to impair chromosome segregation during mitosis. There are some interesting results in this study, including Rb depletion inducing centromeric transcription, Rb depletion inducing DNA damage at centromeres or centromere-proximal regions. However, there are also some claims that are not well supported by their evidence. I listed a few specific points as follow.

1. Rb functions in various cellular processes including cell cycle progression, differentiation, and apoptosis. Its role in transcriptional regulation is mainly through E2F. Its depletion could affect the expression of a wide range of genes. Therefore, it is not clear whether the observed phenotypes are a direct consequence of Rb depletion on centromeric transcription. It seems that there is no mentioning or discussion of this at all by the authors in the paper. I believe that further testing it would help uncover the underlying mechanisms. At least, one way to test the possibility of directness is to examine if Rb protein could localize to centromeres using immunofluorescence.

2. The authors claimed that Rb depletion causes DNA damage through increased accumulation of centromeric R-loops that are derived from the enhanced centromeric transcription. Although it is an interesting point, the evidence to support it may not be sufficient. In the paper, there are not any results showing changes of R-loops in response to Rb1 depletion. DRIP is an extensively used approach to examine the presence of R-loops on chromatin. It is also sensitive enough to quantify R-loops. The presence of R-loops at centromeres has also been demonstrated by various previous studies. Therefore, it is feasible to examine the changes of R-loops using DRIP-PCR with the centromere primers used in this paper.

3. An alternative possibility can also explain the claim made by the authors for the relationship of DNA damage and centromeric transcription. Recent two publications demonstrated that, in response to DNA damage, especially double-strand breaks, the transcription of alpha-satellite, a major type of tandemly repeated DNA sequence in human centromeres, is dramatically increased (Teng et al., 2024, Nature Communications; Yilmaz et al., 2021, Nature). Given the finding that RB depletion can lead to DNA damage through several mechanisms, it is therefore possible that RB depletion-caused DNA damage can dramatically induce centromeric transcription. I think that it is important to discuss, at least, these findings in this manuscript, as they may represent an alternative explanation for the authors' findings.

4. In Figure 5A, in the sample of Ind-cenSuv/siScr, it seems that this cell suffers cohesion defects to some degree, as revealed by (partially) separated sister centromeres (ACA). If this is the case, the decreased Aurora B in this condition could also be derived from impaired centromeric cohesion and displaced Sgo1, which could be another explanation for the observed lagging chromosome in the paper. I think that to repeat this experiment with chromosome spread could clarify if there is indeed centromeric cohesion defects, at least partially, under this condition.

5. The data regarding Suv420H2 targeting to centromeres are intriguing. However, the authors should further substantiate this to the several changed signals in RB depletion cells, including γ H2AX, p-ATR, and RNAPII Ser2 phosphorylation.

6. Figure 2E and 4B are missing important controls. The authors should include a negative control of no EU to validate the specificity and efficacy of nascent RNA labeling. In addition, the y-axis in Figure 4B should include a maximum scale value for clarity.

7. There appears to be a labeling error for γ H2AX and ACA in Figure 2C.

We thank the editor and reviewers for their careful and enthusiastic review of our manuscript. We have carefully considered their feedback and have now added additional experimental data to address the major concerns that were raised. These new experiments include orthogonal approaches to study R-loop formation and additional assessment of RNAPII pS2 localization at centromeres. Using DRIP-qPCR experiments, we now directly demonstrate that the presence of R-loops at mitotic centromeres increases following RB loss and that the observed increase in R-loops is dependent on RNAPII activity. In addition, we now also show that Suv420H2 localization at centromeres suppresses RNAPII activity (RNAPII pS2) and γ H2AX foci formation in RB depleted mitotic cells, complementing experiments in the original manuscript that show molecular tethering of Suv420H2 to centromeres is sufficient to reduce levels of centromere proximal breaks and suppress mitotic defects that arise following RB loss. We have corrected minor errors in the text and figures. In-text changes to introduce these new data and to expand upon additional points raised by reviewers appear in blue font. Please see below for a point-by-point response to reviewer comments:

Reviewer #1 (Comments to the Authors (Required)):

In this manuscript Crowley et al. investigate the role of centromeric transcription in maintaining mitotic fidelity and its perturbation following the loss of the tumor suppressor RB. Building on a growing body of literature that highlights the necessity of RNAPII-mediated transcription at centromeres for proper CENP-A deposition and kinetochore assembly, the authors demonstrate that RB depletion leads to elevated transcription at both centromeric and pericentromeric regions. This transcriptional upregulation correlates with increased recruitment of pATR and Aurora B, factors associated with replication stress and mitotic surveillance. Broadly, we found the premise of this work very interesting, and in line with growing evidence in the field linking cancer pathways to regulation of CENP-A deposition. Congratulations on a nicely executed study.

Three broad points the authors might wish to consider:

1) This study presents compelling evidence of centromere dysfunction, particularly relevant in cancer cells where RB is frequently mutated or functionally inactivated. However- it will be beneficial to look at actual cancer cells where Rb is mutated to determine the pathological significance. We accept that this may be beyond the scope of the current study, but maybe discuss this a bit further?

We share this reviewer's excitement in this possibility and are eager to examine the extent to which centromere dysfunction is compromised in RB deficient (or inactivated) cancer cells. However, the extensive experimentation needed to explore this possibility is beyond the scope of the current study.

2) the authors propose that RB loss promotes R-loop formation at these repetitive loci, which in turn drives aberrant transcription and genome instability. This is where the manuscript could use a bit of more direct evidence for R-loop formation at centromeres or pericentromeres in RB-deficient cells. Incorporation of R-loop mapping (e.g., DRIP-seq) would significantly strengthen this model significantly and hugely elevate the importance and impact of this work.

We have now completed DRIP-qPCR experiments to isolate and perform comparative measures of R-loops at centromeres in mitotic cells with and without RB depletion. We have completed these experiments in the presence and absence of the RNAPII inhibitor α -amanatin. Consistent with the model proposed in the original manuscript, we find that R-loops are increased at the centromeres of mitotic cells that lack RB and that the increase in R-loops is dependent on RNAPII activity. This new data can be found **Supplemental Figure 1B**.

3) the authors also address the pathological consequences of dysregulated centromere transcription. The authors highlight that excessive expression of centromeric and pericentromeric repeats, often observed in cancer, leads to the formation of R-loops-structures that trigger replication stress, activate ATR kinase, and ultimately enhance Aurora B activity. While Aurora B is necessary for error correction, its hyperactivation, particularly in mitosis, can paradoxically promote chromosome segregation errors and genome instability. Again, this is a bit of a weak link, as the mechanistic pathway linking R-loop formation to Aurora B activation, however, remains to be clearly defined. Although the study centers on centromere integrity, the authors do not report on key inner kinetochore components such as CENP-A or CENP-C, whose misregulation could provide further insight into mitotic errors observed in RB-deficient cells especially in the context of increased transcription. We encourage the authors to consider adding such data that we think would make this work a whole lot more insightful.

We now include immunofluorescence experiments that examine CENPA intensity in mitotic cells. We find no difference in CENPA levels at centromeres in control and RB-depleted conditions. These new data can be found in **Supplemental Figure 3B & C**. Importantly, we note that since CENPA deposition is cell-cycle regulated (Reviewed in Rowley and Jansen, 2025), transcription-dependent changes in CENPA deposition may require more than one cell cycle to become evident. Thus, while we do not see CENPA changes following acute RB depletion (48h siRNA treatment: analyzed cells are likely in their 1st mitosis following RB loss), constitutive perturbation of centromere transcription could have a more profound impact on CENPA levels at mitotic centromeres.

Other experimental or technical points to consider:

1. In Figure 1 the authors show primarily by immunofluorescence that RNAP II pS2 is upregulated at centromeres upon Rb depletion. A better way to quantitatively assay increased RNAPII pS2 upon depletion of RB is to perform ChIP qPCR at the centromeric DNA for both RNA pol II and RNAPII pS2. This can be incorporated in the manuscript.

We agree that ChIP qPCR is a robust approach to assess RNAPII and RNAPII pS2 localization in interphase cells. However, time constraints and expense of performing these experiments in isolated mitotic cells (which represent only ~2% of the asynchronous population of proliferative RPE cells) is prohibitive. Indeed, our immunofluorescence assays are in line with other studies that explore RNAPII localization in mitotic cells (Chan et al., 2012) and those demonstrating that ChIP-qPCR analysis of mitotic RNAPII pS2 mirrors immunofluorescence analysis of the same (Perea-Resa et al., 2020).

2. R-DNA loops can form also by RNA pol II pausing (Xu, C., Li, C., Chen, J. et al. Nature 621, 610-619 (2023). <https://doi.org/10.1038/s41586-023-06515-5>). The authors should delineate in this study that whether there is RNAP II pausing or not at the centromeres

Phosphorylation of S2 on RNAPII is a mark associated with elongation. Our interpretation that RB deficient cells exhibit an increase in active/elongating RNAPII is consistent with observations that centromere-localized RNAPII pS2 levels are increased. However, as reviewer 1 points out, we have not assessed the centromere level of RNAPII pS5, a mark of paused polymerase and therefore can not discount that paused polymerase is contributing to R-loop formation. We have reviewed and revised the text to ensure that discussion of our 'rescue' experiments (showing that RNAPII inhibition reduces R loop formation, DNA damage and mitotic errors) focuses on overall inhibition of RNAPII activity and does not otherwise imply that elongation (and not pausing) is the source of R-loop formation.

3. In Figure 1E the authors measure the nascent RNA levels in absence of RB. The authors should also determine the expression of LINEs and SINEs elements at the centromeres or pericentromeres.

We now include data showing that representative LINE transcripts are also increased following RB loss. These new data can be found in **Supplemental Figure 1A**.

4. The authors although later in the manuscript comment on the link between RB and recruitment of certain epigenetic enzymes, however it will be better to show the link at the endogenous level. What happens to H4K20me3, H3K9me3 at the centromeres when cells are depleted of the RB protein?

The relationship between RB and Suv420H2 is well-described. RB has been demonstrated to directly interact with Suv420H2 (Gonzalo et al., 2005), and the loss of RB to dramatically reduce H4K20 methylation (Gonzalo et al., 2005, Isaac et al., 2006, Manning et al., 2014). Our 2014 study in *Molecular Cell* used ChIP experiments to demonstrate that endogenous levels of Suv420H2 and its corresponding methyl mark (H4K20me3) are reduced at pericentromeres following RB depletion. **These published studies are highlighted in the results and discussion sections.**

5. In figure 2C the labelling of H2AX and ACA is not proper. The authors should be careful in labelling the figures as that might lead to erroneous interpretation of the data.

We thank the reviewer for catching this error- **the figure has been revised to correctly label the stained panels.**

6. In Figure 2E the authors employed a fluorescent in situ hybridization (FISH) approach, exoFISH. To confirm the formation of R-DNA loops the authors should use other direct methods like DRIP seq or using GFP-catalytically dead RNase H1 for visualization at the centromeres in siRB cells.

We have now included DRIP-qPCR experiments to verify R-loops at mitotic centromeres. This data is presented in new **Supplemental Figure 1B** (see also response to reviewer 1 major critique #2)

7. Also, another way to relieve R-DNA loop formation is treating cells with Topoisomerase inhibitors. The authors should also assay the transcription at the centromeres upon these inhibitors in the Rb deficient cells to confirm that it is indeed through the R-DNA hybrids

To address this suggestion, we attempted experiments treating mitotic-arrested control and RB-deficient cells with topoisomerase inhibitors. Unfortunately, we find that etoposide treatment is incompatible with our established staining protocol and we are therefore unable to address this concern.

8. The authors also do not show how the enhanced transcription at the centromere or pericentromeres affect CENP-A/C deposition as they are both intricately co regulated.

We have now examined and find no impact of acute RB depletion on CENPA centromere levels in mitotic cells. This new data is presented in **supplemental figure 3C and D**. Please also see response to reviewer 1, major critique #1 above.

9. In figure 3A and 3B the authors show that in absence of RB the pATR is high, when ATRi the laggards are decreased. However, the authors do not comment on the R DNA loop formation upon ATRi, does it decrease also?

We have now performed DRIP-qPCR in the presence of ATRi and find that inhibition of ATR activity does not rescue the increased R-loop formation found at centromeres of mitotic cells depleted of RB. These data are consistent with ATR acting downstream/in response to R-loop formation. This new data can be found in **Supplemental Figure 4A**.

10. RB (retinoblastoma protein) and E2F pathway plays a crucial role in regulating the cell cycle, particularly the transition from G1 phase to S phase. Do the authors here characterize the cell cycle of the Rb deficient cells? Also for the mitotic defects are laggards the only mitotic defects they observe in the Rb deficient cells?

We and others have previously characterized the impact of RB loss on cell cycle progression and find that, in highly proliferative cells (like hTERT-RPE-1 cells) RB loss does not alter overall cell cycle timing (Amato et al., 2009; Manning et al., 2010). Although a small number of chromatin bridges and acentric chromatin (broken chromosomes) are observed following RB loss, the predominant anaphase defect in hTERT-RPE-1 cells is the presence of lagging chromosomes (Manning et al., 2010, Zamalloa et al., 2023).

11. In figure 4 and 5 the authors show that tethering the Suv420H2-GFP at the centromeres reduced the ExoIII-dependent cenFISH signal and the nascent RNA transcription. Subsequently a reduction of AurKB at the centromeres. A direct interaction of Suv420 and RB at centromeres are not shown. Also, in siRB cells does the Suv420 endogenous levels change at the centromeres?

Previous work from several groups has demonstrated physical interaction between Suv420H2 and RB (Gonzalo et al., 2005; Sanidas et al., 2019), and shown that depletion of RB reduces Suv420H2 levels at the centromere (Gonzalo et al., 2005, Isaac et al., 2006, Manning et al., 2014). **These points are highlighted in the results and discussion sections.** (see also response to reviewer 1 minor point #4)

12. RB is often mutated in cancer cells, validating this model in any cancer cells harboring RB mutations is a good model to validate the findings of this study.

We agree with the reviewer on this point, however we believe that the extensive experimentation needed to explore this possibility is beyond the scope of the current study.

Reviewer #2 (Comments to the Authors (Required)):

RB deficiency is associated with chromosomal instability. Previous studies, including work by the authors, demonstrated that this instability can be partly attributed to defects in chromosome segregation. While these defects were linked to deregulation of heterochromatic marks and altered cohesin binding at pericentromeric regions, the precise mechanism by which RB safeguards chromosome segregation fidelity remained unclear.

In this study, the authors conducted a detailed analysis revealing that RB epigenetically regulates transcriptional activity at centromeres, thereby modulating ATR and AurB kinase activity to prevent mitotic errors. Specifically, they demonstrate that:

- RB depletion increases RNAPoIII activity at mitotic centromeres.
- Mitotic RB-deficient cells show a marked increase in newly synthesized centromeric and pericentromeric transcripts.
- RB-deficient cells exhibit a higher proportion of mitotic cells with DNA damage.
- Inhibition of RNAPoIII with α -amanitin in RB-deficient cells restores DNA damage levels to those seen in control cells.
- Centromeres in RB-deficient cells are more sensitive to exonuclease digestion.
- ATR activity and recruitment of AurB kinase at mitotic centromeres are elevated upon RB loss.
- ATR activation and AurB localization are dependent on centromeric transcription.
- Anaphase defects in RB-deficient cells are suppressed by inhibition of either RNAPoIII or ATR.
- Tethering Suv420H2, an RB-bound histone methyltransferase, to centromeres reduces RB loss-induced transcriptional upregulation at mitotic centromeres.
- Suv420H2 tethering also decreases centromeric DNA damage in RB-deficient mitotic cells.
- Finally, Suv420H2 recruitment to centromeres reduces AurB localization and anaphase defects in RB-deficient cells.

The proposed mechanism reveals a previously unrecognized role of RB during mitosis and provides important insights into the consequences of RB loss in tumors, particularly

those marked by genomic instability. A key observation, which is appropriately emphasized in the discussion, is that the rescue strategies used to mitigate segregation defects in RB-null cells specifically target mitosis (e.g., RNAPolIII inhibition in nocodazole-arrested cells) and centromere-localized epigenetic regulation (via cen-Suv420H2 expression). These findings suggest that mitotic errors following RB loss arise primarily from dysregulated centromeric transcription late in the cell cycle, rather than from canonical RB-mediated repression of E2F target genes. This supports a novel, non-canonical role for RB in safeguarding mitotic fidelity.

Although the timing of Suv420H2 recruitment to centromeres remains undefined, the authors could consider discussing whether this function reflects a mitosis-specific activity of RB or could involve earlier, interphase-mediated recruitment mechanisms. Overall, the manuscript is well written, and the authors' conclusions are well supported by the data. I co-reviewed this manuscript with a postdoctoral fellow in my lab, and we both strongly recommend it for publication.

We thank the reviewer for the kind comments and thoughtful critique. Published studies demonstrate that Suv420H2 can bind to phosphorylated RB, suggesting an interaction at later stages of the cell cycle (Isaac et al. 2006; Longworth and Dyson, 2010), but do not clarify if this interaction is restricted to G2 or may persist into mitosis. In response to this feedback, we have expanded the text to include discussion of the timing during which RB would likely interact with/regulate Suv420.

Minor comment:

The catalog numbers for the commercially available primary antibodies used in this study are missing and should be included for reproducibility.

We now include the catalog numbers for all antibodies within the methods section.

Reviewer #3 (Comments to the Authors (Required)):

This study tried to build up a story in which Rb depletion results in mitotic defects via centromeric transcription upregulation. Specifically, the authors claimed that Rb depletion upregulates centromeric transcription during mitosis, which increases the amount of R-loops that lead to DNA damage at centromeres. DNA damage then activates ATR at centromeres, which recruits more Aurora B to impair chromosome segregation during mitosis. There are some interesting results in this study, including Rb depletion inducing centromeric transcription, Rb depletion inducing DNA damage at centromeres or centromere-proximal regions. However, there are also some claims that are not well supported by their evidence. I listed a few specific points as follow.

1. Rb functions in various cellular processes including cell cycle progression, differentiation, and apoptosis. Its role in transcriptional regulation is mainly through E2F. Its depletion could affect the expression of a wide range of genes. Therefore, it is not clear whether the observed phenotypes are a direct consequence of Rb depletion on centromeric transcription. It seems that there is no mentioning or discussion of this at all by the authors in the paper. I believe that further testing it would help uncover the underlying mechanisms. At least, one way to test the possibility of directness is to examine if Rb protein could localize to centromeres using immunofluorescence.

We thank the reviewer for their careful critique and interest in our study.

Although centromere enrichment of RB is not apparent at centromeres by immunofluorescence, published ChIP studies have described RB enrichment at centromeric heterochromatin (Ishak et al., 2016) and demonstrated a critical role for RB in recruiting/regulating epigenetic modifiers such as EZH2 and Suv420H2 to centromeres in G2/M (Gonzalo et al., 2005, Manning et al., 2014, Ishak et al., 2016). Together, these studies demonstrate that RB is at the 'right place' and functioning to regulate G2/M centromere heterochromatin in preparation for mitotic chromosome segregation.

While we can not rule out the possibility that RB-dependent gene expression early in the cell cycle contributes to centromere dysregulation later in the cell cycle (for example via upregulation of an RNAPII activator), we have carefully designed our experiments to specifically perturb transcription and assess the consequences *within a single mitosis*, affording us the temporal resolution to suggest that the centromere damage, ATR activity, AurB enrichment, and anaphase defects are responsive to mitotic transcription (and not merely transcription earlier in the cell cycle, which is not perturbed in our rescue experiments). Similarly, our molecular tethering experiments to enhance repressive marks and suppress transcription at the centromere (and not elsewhere in the genome) affords us the spatial resolution to strongly suggest that dysregulation of centromeres (and not other RB-regulated regions of the genome) are the driving force in the observed mitotic defects. These points are highlighted in the results and discussion sections.

2. The authors claimed that Rb depletion causes DNA damage through increased accumulation of centromeric R-loops that are derived from the enhanced centromeric transcription. Although it is an interesting point, the evidence to support it may not be sufficient. In the paper, there are not any results showing changes of R-loops in response to Rb1 depletion. DRIP is an extensively used approach to examine the presence of R-loops on chromatin. It is also sensitive enough to quantify R-loops. The

presence of R-loops at centromeres has also been demonstrated by various previous studies. Therefore, it is feasible to examine the changes of R-loops using DRIP-PCR with the centromere primers used in this paper.

We thank reviewer 3 for this suggestion and now include DRIP-qPCR experiments that demonstrate an increase in centromeric R-loops following RB depletion. This new data can be found in Supplemental Figure 1B.

3. An alternative possibility can also explain the claim made by the authors for the relationship of DNA damage and centromeric transcription. Recent two publications demonstrated that, in response to DNA damage, especially double-strand breaks, the transcription of alpha-satellite, a major type of tandemly repeated DNA sequence in human centromeres, is dramatically increased (Teng et al., 2024, Nature Communications; Yilmaz et al., 2021, Nature). Given the finding that RB depletion can lead to DNA damage through several mechanisms, it is therefore possible that RB depletion-caused DNA damage can dramatically induce centromeric transcription. I think that it is important to discuss, at least, these findings in this manuscript, as they may represent an alternative explanation for the authors' findings.

We thank the reviewer for raising this alternative hypothesis. We have tried to clarify the relationship between centromere transcription and DNA damage in our system by systemically perturbing one step in the model and analyzing the proposed upstream and downstream steps. If DNA damage were to promote alpha satellite transcription in our system, we would expect the RNAPII inhibitor α -amanitin to reduce transcription without rescuing levels of centromere damage. However, our data do not support this model. Instead, we find that inhibition of mitotic centromere transcription (via α amanitin treatment within mitosis or centromere tethering of the epigenetic repressor Suv420H2) rescues DNA damage at centromeres, thereby better supporting a model where DNA damage is downstream of transcription (transcription \rightarrow R loops \rightarrow DNA damage; Figures 2 & 3, Supplemental Figure 1). **We now include discussion of this alternative hypothesis in our results section.**

4. In Figure 5A, in the sample of Ind-cenSuv/siScr, it seems that this cell suffers cohesion defects to some degree, as revealed by (partially) separated sister centromeres (ACA). If this is the case, the decreased Aurora B in this condition could also be derived from impaired centromeric cohesion and displaced Sgo1, which could be another explanation for the observed lagging chromosome in the paper. I think that to repeat this experiment with chromosome spread could clarify if there is indeed centromeric cohesion defects, at least partially, under this condition.

We do not observe any evidence of cohesion fatigue or cohesion defects in our cenSuv/siScr cells (including with chromosome spreads, as described in Herlihy et al., 2021). To avoid confusion, we have replaced the representative image of this condition.

5. The data regarding Suv420H2 targeting to centromeres are intriguing. However, the authors should further substantiate this to the several changed signals in RB depletion cells, including γ H2AX, p-ATR, and RNAPII Ser2 phosphorylation.

We thank the reviewer for the suggestion and now include experiments showing γ H2AX, RNAPII total, and RNAPII pSer2 in our cen-Suv420 system. Consistent with our described repression of centromere transcription, we find that tethering of Suv420H2 in RB deficient cells restores active RNAPII (pS2) and γ H2AX at mitotic centromeres to levels seen in control cells. This new data can be found in **Supplemental figure 5B-E and Supplemental Figure 6A & B.**

6. Figure 2E and 4B are missing important controls. The authors should include a negative control of no EU to validate the specificity and efficacy of nascent RNA labeling. In addition, the y-axis in Figure 4B should include a maximum scale value for clarity.

The click-IT reaction that links EU-labelled RNA to the biotin beads is highly specific and the washes quite stringent. We are not aware of a study in which a non-EU negative control is used for comparison. For our analysis of nascent RNA, we have included a total RNA control that represent the quantification of non-EU labelled RNA molecules, this is on par with recently published work employing similar approaches Chen et al 2021, JCB; Chen et al., 2024, eLife; Flynn et al., 2011, PNAS; Palozola et al., 2017, Science. We have updated the y-axis in Figure 4B.

7. There appears to be a labeling error for γ H2AX and ACA in Figure 2C.

We apologize for the oversight and have corrected this error.

December 2, 2025

RE: Life Science Alliance Manuscript #LSA-2025-03433-TR

Dr. Amity L Manning
Worcester Polytechnic Institute
Biology and Biotechnology
60 Prescott St
Worcester, MA 01605

Dear Dr. Manning,

Thank you for submitting your revised manuscript entitled "RB dependent transcriptional regulation at mitotic centromeres preserves genome stability". Your revised manuscript was evaluated by all the original reviewers whose comments are appended below.

As you will read, the three reviewers are satisfied that the revised manuscript has addressed all their concerns. In line with the reviewers' evaluation, we would be happy to publish your paper in Life Science Alliance pending final revisions necessary to meet our formatting guidelines.

- For description of imaging in the methods, please provide numerical aperture for objectives.
- Please provide details for all antibodies used in immunofluorescence (for example: antibodies against GFP, Aurora B)
- Please specify if scale bars apply to multiple figure panels in the legends (for example in Figure 2, Figure 5, Figure S3). Please define the scale bar for Figure S2.
- Please include a statement on Data availability following LSA guidelines, <https://www.life-science-alliance.org/editorial-policies#data-sharing>
- Please add the X and Bluesky handles of your host institute/organization, as well as your own and/or one of the authors, in our system
- Please use the [10 author names, et al.] format in your references (i.e., limit the author names to the first 10)
- We encourage you to revise the figure legends for figures 5 and S5 such that the figure panels are introduced in alphabetical order
- Figure S4 has only one panel; therefore, please remove the label A from the current figure and its legend
- Please add callouts for Figures S5A-E and S6A-D to your main manuscript text
- Please be sure that the authorship listing and order is correct

A. FINAL FILES:

- An editable version of the final text (.DOC or .DOCX) is needed for copyediting (no PDFs).
- High-resolution figure, supplementary figure and video files uploaded as individual files: See our detailed guidelines for preparing your production-ready images, <https://www.life-science-alliance.org/authors>
- Summary blurb (enter in submission system): A short text summarizing in a single sentence the study (max. 200 characters)

including spaces). This text is used in conjunction with the titles of papers, hence should be informative and complementary to the title. It should describe the context and significance of the findings for a general readership; it should be written in the present tense and refer to the work in the third person. Author names should not be mentioned.

B. MANUSCRIPT ORGANIZATION AND FORMATTING:

Thank you for your attention to these final processing requirements. Please revise and format the manuscript and upload materials as soon as you are able.

Sincerely,

Sarita Hebbar, PhD
Scientific Editor
Life Science Alliance
<http://www.lsajournal.org>

Reviewer #1 (Comments to the Authors (Required)):

The authors have provided reasonable responses to queries raised and incorporated edits that address issues raised.

We think the work is ready for publication- congratulations to all the authors on a nicely executed and important study! -Yamini Dalal & Sweta Sikder (NCI/NIH).

Reviewer #2 (Comments to the Authors (Required)):

This is an improved version of an already strong manuscript.

The addition of the DRIP-qPCR experiments that directly demonstrate induction of R-loop formation at mitotic centromeres following RB loss, and that this effect is RNAPoIII-dependent and ATRi-independent, substantially strengthens the proposed model.

Reviewer #3 (Comments to the Authors (Required)):

Thanks for the author's time and efforts on the revisions. I am satisfied with the response.

December 11, 2025

RE: Life Science Alliance Manuscript #LSA-2025-03433-TRR

Dr. Amity L Manning
Worcester Polytechnic Institute
Biology and Biotechnology
60 Prescott St
Worcester, MA 01605

Dear Dr. Manning,

Thank you for submitting your Research Article entitled "RB dependent transcriptional regulation at mitotic centromeres preserves genome stability". It is a pleasure to let you know that your manuscript is now accepted for publication in Life Science Alliance. Congratulations on this interesting work.

Your manuscript will now progress through copyediting and proofing. At the proofing stage, please remove the current statement under data availability as it refers to availability of materials in this work. Instead, please include a statement that refers to the data underlying results in this work. Please feel free to use this "Raw data underlying this work are available from the corresponding author upon reasonable request."

DISTRIBUTION OF MATERIALS:

Again, congratulations on a very nice paper. I hope you found the review process to be constructive and are pleased with how the manuscript was handled editorially. We look forward to future exciting submissions from your lab.

Sincerely,

Sarita Hebbar, PhD
Scientific Editor
Life Science Alliance
<http://www.lsajournal.org>